# Green and Traditional Synthesis of Copper Oxide Nanoparticles—Comparative Study

**DOI:** 10.3390/nano10122502

**Published:** 2020-12-14

**Authors:** Obakeng P. Keabadile, Adeyemi O. Aremu, Saheed E. Elugoke, Omolola E. Fayemi

**Affiliations:** 1Department of Chemistry, Faculty of Natural and Agricultural Sciences, North-West University (Mafikeng Campus), Private Bag X2046, Mmabatho 2735, South Africa; 29582172@student.g.nwu.ac.za (O.P.K.); elugokesaheed@gmail.com (S.E.E.); 2Material Science Innovation and Modelling (MaSIM) Research Focus Area, Faculty of Natural and Agricultural Sciences, North-West University (Mafikeng Campus), Private Bag X2046, Mmabatho 2735, South Africa; 3Indigenous Knowledge Systems Centre, Faculty of Natural and Agricultural Sciences, North-West University (Mafikeng Campus), Private Bag X2046, Mmabatho 2735, South Africa; Oladapo.Aremu@nwu.ac.za

**Keywords:** green copper oxide, nanoparticles, spectroscopy, cyclic voltammetry, terminalia phanerophlebia

## Abstract

The current study compared the synthesis, characterization and properties of copper oxide nanoparticles (CuO) based on green and traditional chemical methods. The synthesized CuO were confirmed by spectroscopic and morphological characterization such as ultraviolet-visible (UV-vis) spectroscopy, fourier transform infrared (FTIR) spectroscopy, zeta potential, scanning electron microscopy (SEM) and energy dispersed X-ray (EDX). Electrochemical behavior of the modified electrodes was done using cyclic voltammetry (CV) in ferricyanide/ferrocyanide ([Fe(CN)_6_]^4−^/[Fe(CN)_6_]^3−^) redox probe. As revealed by UV spectrophotometer, the absorption peaks ranged from 290–293 nm for all synthesized nanoparticles. Based on SEM images, CuO were spherical in shape with agglomerated particles. Zeta potential revealed that the green CuO have more negative surface charge than the chemically synthesized CuO. The potential of the green synthesized nanoparticles was higher relative to the chemically synthesized one. Cyclic voltammetry studies indicated that the traditional chemically synthesized CuO and the green CuO have electrocatalytic activity towards the ferricyanide redox probe. This suggests that the green CuO can be modified with other nanomaterials for the preparation of electrochemical sensors towards analytes of interest.

## 1. Introduction

Nanotechnology generally involves the application of extremely small particles that are used across all fields of science including chemistry, biology, medicine and material science [1,2,3,4]. According to Sathiyavimal et al. [5], nanotechnology deals with the synthesis of metal and metal oxide nanoparticles of different sizes, shapes, disparity and chemical composition. Nanoparticles, which by definition are the clusters of atoms in the size range of 1–100 nm are the major building blocks of nanotechnology [6,7].

There are two methods of synthesizing nanoparticles, which are the chemical and physical methods [8]. The chemical method of synthesis includes chemical reduction, electrochemical techniques and photochemical reduction [9,10]. The classical chemical method in which a reducing agent such as sodium borohydride and hydrazine [6] as well as radiation chemical method generated by ionization radiation are also used for chemical synthesis of nanoparticles. Even though most chemical methods successfully produce pure and well-defined nanoparticles, they are expensive, inefficient and can release harmful wastes to the environment hence a better eco-friendly methods are preferred [10]. On the other hand, the chemical method of nanoparticles synthesis give room for the use of desired reducing agents with lower economic implication compared to the physical method of synthesis such as the laser ablation technique which attracts extra cost of the laser system procurement.

The physical methods of synthesis include condensation, evaporation and laser ablation [8]. Green synthesis method (plant-mediated synthesis of nanoparticles) is established as an alternative to physical and chemical methods. It is simple, rapid process that involve the use of less toxic and environmentally-friendly materials as compared to other methods of synthesis [7,11]. Green synthesis mitigates environmental problems such as solar conversation, catalysis and agricultural production [12]. According to Zhang et al. [13], the plant material in the green synthesis often have a vital role in the size and surface morphology of the nanoparticle synthesized. The plant extract can act as both the reducing and capping agent. In terms of size of the nanoparticles formed, the products of green synthesis are larger compared to the ones obtained by chemical methods [8]. In addition, nanomaterials produced from the green source have high biocompatibility which contributes to their biofunctionality [14]. It has also been reported that the green synthetic route produces higher yield of nanoparticles than the chemical means [15].

Nanoparticles of a wide range of elements such as copper (Cu) [16], gold (Au) [17], silver (Ag) [18], titanium (Ti) [19], zinc (Zn) [20], silicon (Si) [21] and palladium (Pd) [22] and their oxides are abound in literature. Specifically, copper is a good conductor of heat or electricity and it is cheaper than silver and gold [23]. Copper oxide nanoparticles (CuO) are used as water purifiers, antimicrobials and are also used in batteries, gas sensors, catalysts for different cross-coupling reactions and high temperature superconductors [7,24,25,26,27]. CuO have a high surface-to-volume ratio that makes them very reactive and interact easily with other materials [28]. Moreover, CuO have moderate band gap, good catalytic activity and high optical transparency which further extend their industrial applications [29,30].

Green synthesis of CuO with good physico-chemical properties have been accomplished with the use of microbial precursors such as *Penicillium aurantiogriseum*, *Penicillium citrinum, Penicillium waksmanii* [31] and *Seratia* sp. [32] as reductants. Likewise, leaf extracts from plants such as *Aloe vera* [33], *Psidium guajava* [34], *Abutilon indicum* [35], *Malva sylvetris* [36], *Carica papaya* [37], *Camellia sinensis* [38] and *Ixiro coccinea* [39] have been used for the preparation of CuO. Interestingly, most of the CuO from these green sources have demonstrated noteworthy antimicrobial activity, particle sizes below 100 nm and photocatalytic activity towards the degradation of contaminants were investigated. For instance, Ijaz et al. [33] prepared CuO from the leaf extract of *Abutilon indicum* and copper (II) nitrate trihydrate. The resultant nanoparticles had noteworthy antimicrobial activity against *Escherichia coli*, *Staphyloccus aureus* and *Bacillus subtilis*, excellent oxidation properties towards linolenic acid as well as good photocatalytic effect on the degradation of Acid Black 210 organic dye.

These precedents informed our interest in the preparation of CuO using *Terminalia phanerophlebia* leaf extracts. Green CuO were prepared from the leaf extract of *Terminalia phanerophlebia* obtained using ethanol (eCuO), deionized water (wCuO) and acetone (aCuO) as solvent. We also synthesized CuO through the traditional chemical means (cCuO) for a comparative study with the green CuO. Precisely, comparative analysis of the spectroscopic, morphological and electrochemical properties of the two nanoparticles were investigated. To the best of our knowledge, this is the first time a CuO prepared from this green precursor will be used in a comparative analysis with the chemical counterpart.

## 2. Materials and Methods

### 2.1. Reagents and Materials

Copper sulphate pentahydrate (CuSO_4_.5H_2_O) (99.5%), sodium borohydride (NaBH_4_) (95%), ethanol and acetone were purchased from Sigma-Aldrich, Gauteng, 1600, South-Africa. All reagents were of analytical grade. Leaves of Terminalia phanerophlebia (Family: Combretaceae) were collected from the Botanical Garden, University of KwaZulu-Natal (Pietermaritzburg campus), South Africa and were positively identified by the curator of the Bews Herbarium, University of KwaZulu-Natal, South Africa.

### 2.2. Preparation of Plants Extracts

Extract from the oven-dried leaves of *Terminalia phanerophlebia* was prepared by weighing 2 g of the ground powder and dissolved in 150 mL deionized water, ethanol and acetone, respectively, as previously reported in literature. The three extracts were filtered and used for the synthesis of nanoparticles.

### 2.3. Synthesis of CuO

#### Traditional Chemical Synthesis of CuO

The traditional chemical synthesis was done according to a method reported by Khatoon et al. [6] Briefly, 60 mL of NaBH_4_ was transferred into 250 mL Erlenmeyer flask which was subsequently placed on a hot plate with magnetic stirrer. The resultant solution was heated for 10 min while stirring at 80 °C. Thereafter, 50 mL of 0.02 M CuSO_4_.5H_2_O was added dropwise to the flask and the final solution was heated for 7 min with continuous stirring. The flask was covered with aluminum foil after completion to avoid oxidation. The particles were allowed to settle leaving the solution colorless and copper nanoparticles with green color settled at the bottom of the flask. The solution was kept in a hot air oven for 24 h to convert the aqueous mixture to powder. The dried powder was collected, dissolved in ethanol and distilled water and centrifuged for purification. This purification was done thrice to remove all impurities and dried for further use. The XRD studies were done with a Rontgen PW3040/60 X’Pert Pro diffractometer by Bruker, Hamburg, Germany, from diffraction angles of 10–80 degrees at room temperature.

### 2.4. Green Synthesis of CuO

The green synthesis was carried out as reported by Sathiyavimal et al. [5] Briefly, 30 mL of 0.1 M CuSO_4_.5H_2_O was added to 10 mL of the plant extract in a flask and then the solution was stirred and heated at 90 °C for 5 h. Color change of the solution (from yellowish green to brownish black) indicated the formation of CuO. The solution was kept overnight at room temperature. The CuO obtained were centrifuged and washed twice with distilled water. The nanoparticles were collected and dried in a hot air oven. Similar procedure was applied for the ethanol and acetone plant extract using temperatures of 68 °C and 46 °C, respectively. The purification of the nanoparticles was done with the approach used for the CuO nanoparticles through the traditional chemical synthetic route.

### 2.5. Characterization of CuO

CuO synthesized were characterized by using Uviline 9400UV-vis spectrophotometer (Sl Analytics, Mainz, Germany) at wavelengths ranging from 200 to 700 nm for UV-vis spectroscopy. Opus Alpha-P Fourier-transform infrared spectrophotometer (Brucker Corporation, Billerica, MA, USA) in the range of 4500–400 cm^−1^ was applied for the IR-spectroscopy. Quanta FEG 250 scanning electron microscope (ThermoFisher Scientific, USA) was used to investigate the morphology and the elemental composition via SEM and EDX analysis. In preparation for SEM analysis, clear suspension of the nanoparticles was made in ethanol and subsequently deposited on a mica solid substrate. The accelerating image, brightness, working distance and contrast of the microscope were adjusted to optimum level prior to imaging. Zeta analyzer was used to obtain the surface charge and the long-term stability of the synthesized nanoparticles. This was accomplished with the aid of a Malvern multi-purpose titrator instrument.

### 2.6. Electrochemical Studies

The cyclic voltammetry characterization was done using a potentiostat with a three electrode system. DropSense with inbuilt Dropview 200 software (Metrohm South Africa (Pty) Ltd., Johannesburg, South Africa) was fitted with screen printed carbon electrode (SPC) having inner carbon working electrode (diameter of 4 mm), a silver pseudo-reference (Ag/AgCl) electrode (RE) and carbon counter electrode (CE). SPC was modified with cCuO, wCuO, aCuO and eCuO to obtain the modified working electrode used for the electrochemical studies. SPC electrode was modified by dropping about 0.5 µL of the ultrasonicated CuO nanoparticles samples on the surface of the electrode and air dried for 20 min. Noteworthy, the ultrasonication of the nanoparticles was done at room temperature. Each of the modified working electrodes were characterized using the Fe(CN)_6_]^4−^/[Fe(CN)_6_]^3−^ (10 mM) redox probe in the presence of a phosphate buffer saline (PBS) (0.1 M, pH 7.0) supporting electrolyte. All cyclic voltammetry (CV) measurements were done at a scan rate of 50 mVs^−1^ over a potential window of 1.0–1.2 V.

## 3. Results and Discussion

### 3.1. Spectroscopic and Microscopic Characterization

#### 3.1.1. UV-vis Spectroscopy Analysis

The UV absorption peak of CuO usually shows an absorption peak ranging from 280 nm to 360 nm [19]. As shown in Figure 1a, a UV absorption peak of 289 nm was recorded for CuO synthesized by a traditional chemical method. As an indication of the formation of different sized CuO, the UV absorption peaks observed for the CuO nanoparticles from green synthesis using water, acetone and ethanol as solvent were 291, 291 and 290 nm for wCuO, aCuO and eCuO, respectively (Figure 1b–d). These observed peaks are within the range reported in literature [25,37,39]. The extra broad peak at 410 nm is similar to the peak reported for green mediated CuO by Saif et al., 2016 [40]. The band gap calculated for wCuO, aCuO and eCuO from the UV-visible spectra were 1.25, 1.75 and 1.63 eV, respectively. These band gaps are higher than a value of 1.20 eV reported for CuO which suggest that the green CuO nanoparticles have higher conductivity than the traditional chemically synthesized CuO nanoparticles [41].

#### 3.1.2. FT-IR Analysis

According to Prakash et al. [42], characteristic peaks of IR spectra of is in the range 400–650 cm^−1^. In the FTIR spectra of CuO from the traditional chemical synthesis, Cu-O stretching was observed at 494 and 595 cm^−1^ (Figure 2a). The peaks observed at 3404 cm^−1^ is associated with OH stretching present as a result of the hydroxyl group on the surface of the CuO. The FTIR spectra of wCuO, aCuO and eCuO were presented in Figure 2b–d. In the spectra of wCuO, Cu-O stretching band was observed at 444 and 579 cm^−1^ and OH stretching at 3117 cm^−1^ (Figure 2b). Figure 2c shows a Cu-O stretching at 450 cm^−1^ and OH stretching at 3353 cm^−1^. In Figure 2d, the absorption peak situated at 450 cm^−1^ corresponds to the Cu-O stretching and the one at 3353 cm^−1^ is due to the OH stretching vibration. Absorption peaks at 1696 cm^−1^ which appeared only in the spectra of wCuO (Figure 2b) is associated with the stretching vibrations of C=C of a water-soluble unsaturated component of the plant extract. As shown in Figure 2, the OH stretching band for a green method are broader when compared to that of the traditional chemical method. This could be due to the hydroxyl group present in the plant extract attached to the CuO.

#### 3.1.3. Energy Dispersed X-ray and SEM Analysis

The dominance of copper and oxygen in all the plots confirms that the synthesized nanoparticles are indeed that of copper oxide (Figure 3. The presence or origin of sodium is a result of the reducing agent that was used for the chemical synthesis (Figure 3a). The percentage of Cu is lower in cCuO and also in eCuO. The standard deviation for the percentage composition of copper in the nanoparticles are 1.27, 4.87, 3.19 and 1.84 for cCuO, wCuO, eCuO and aCuO, respectively.

This confirms the action of ethanol as co-surfactant in the preparation of CuO from plant extract using ethanol as solvent as suggested by the SEM micrograph. Higher percentage of oxygen was evident in all the green mediated copper nanoparticles (Figure 4b–d) relative to the traditional chemical synthesis. The standard deviation for the percentage composition of oxygen in the nanoparticles are 0.87, 4.37, 2.0 and 1.84 for cCuO, wCuO, eCuO and aCuO, respectively. This can be attributed to the high oxygen content of the oxygen containing functionalities in the plant extract.

#### 3.1.4. X-ray Diffraction (XRD) Studies

The XRD diffractogram gives an insight to the crystalline structure of a compound. Crystallite size was calculated using X-ray diffraction pattern of CuO nanoparticles (Figure 5). Diffraction peaks for chemically synthesized cCuO (Figure 4a) were noticed at 2θ values of Brags angle for 22.85° (020), 28.00° (021), 30.58° (110), 33.42° (002) 35.67° (111), 41.43° (131), 52.45° (113) and 59.16° (200) and for aqueous extract of plant (Figure 5B) at 22.9° (020), 27.96° (021), 30.60° (110), 33.44° (002) 35.67° (111), 41.34° (131), 52.69° (113) and 60.1° (200) with their corresponding lattice/Miller indices in parentheses. XRD pattern of CuO synthesized from plants extract via acetone and methanol aCuO and eCuO were deficient of peaks (graphs not shown) which possibly could have been from impurities from the organic solvent used in the preparation of the extract. Inter planar d-spacing of cCuO and wCuO were calculated applying Bragg’s law equation: [43].
(1)2dSinθ=nλ
where d is interplanar spacing, θ is Brag’s angle of diffraction = 2θ/2, λ is X-ray wavelength (0.154 nm) and n = 1 as represented in Table 1.

Crystallite size calculated from the diffraction peak corresponding to most intense peak (θ/2) for cCuO (22.85°) and wCuO (22.9°) using Debye–Scherrer’s formula were found to be 17.8 nm and 24 nm for cCuO and wCuO, respectively, which are close to reported values in the range of 14–25 nm. [44,45] Differences in the size could, possibly, be attributed to different route of synthesis, imperfect crystallization and strain.

### 3.2. Zeta Potential Studies

The size of the zeta potential gives information about the particle synthesized stability. For example, particles with high zeta potential exhibit increased stability due to larger electrostatic repulsion between particles [46]. While cCuO show a charge of −5.60 mV, wCuO, aCuO and eCuO had corresponding charges of −22.3, −16.5 and −15.6 mV, respectively (Figure 6). All the CuO synthesized had negative surface charges. As highlighted by Aparna et al., [47] negative zeta potential obtained for nanoparticles might be an indication that CuO are negative particles and moderately stable. The obtained zeta potentials for green synthesis method were about three times greater than that of traditional chemical method which suggests that CuO synthesized by green method are more stable than those from chemical method. Higher zeta potentials were obtained from the CuO prepared from wCuO (Figure 6b) and aCuO (Figure 6c) compared to the cCuO (Figure 6a) and eCuO (Figure 6d). This suggests lesser agglomeration of the particles which manifested in a better porosity as seen in Figure 4b–c. Zeta potential close to that of wCuO (−28.9 mV) has earlier been reported by Sankar et al. [37] for green CuO prepared from the leaf extract of *Carica papaya*.

### 3.3. Electrochemical Study Using Cyclic Voltammetry

Electrochemical comparative studies of CuO was studied using cyclic voltammetry at a scan rate of 50 mV/s in 10 mM [Fe(CN)_6_]^4−^/[Fe(CN)_6_]^3−^ solution prepared in 0.1 M PBS. The screen print carbon electrode was modified with the CuO nanoparticle synthesized from chemical and the green method. The screen print electrodes were modified with the CuO nanoparticles and denoted as SPC/cCuO, SPC/wCuO, SPC/aCuO and SPC/eCuO, representing chemically prepared CuO nanoparticles and the green mediated CuO from water, acetone and ethanol solvents, respectively (Figure 7). The voltammogram of the modified SPC electrodes of SPC/cCuO, SPC/wCuO, SPC/aCuO and SPC/eCuO showed redox peaks, an anodic peak at about 0.25 V and cathodic peaks at reduction potential of 0.6 V for the [Fe(CN)_6_]^4−^/[Fe(CN)_6_]^3−^ probe. Another observable anodic peak ranging from observed 1.0 V to 1.5 V in Figure 7 is attributed to the CuO on the SPC electrodes. The SPC/cCuO gave better current response compared to the SPC/wCuO, SPC/aCuO and SPC/eCuO modified electrodes (Figure 7) which can be attributed to its lower band-gap between the valence and the conduction band as seen in the UV study also the XRD showed that cCuO has smaller particle size as compared to that of the green mediated CuO nanoparticles. This better current response remained conspicuous with varying scan rates as evident in Figure 8a–d. This could be ascribed to the better catalytic activity of the chemically synthesized CuO towards the redox probe. The anodic peak current of the CuO follows the order; SPC/cCuO > SPC/wCuO > SPC/eCuO > SPC/aCuO. This result confirms that chemical synthesis enhances the electroactivity of the synthesized nanoparticles better than the green mediated nanoparticle. However, the electroactivity of the green mediated CuO can be enhanced by modification with conducting materials such as carbon nanotubes, quantum dots and graphene or graphene oxide for enhance electroactivity.

The effect of the scan rate was also studied for different CuO that are synthesized as seen in Figure 8a–d. The results indicated a direct proportional relationship between scan rate and the current. Increase in scan rate resulted in an increase in current. Linear plots for the comparative are shown in Figure 9 with regression values of approximately 0.9, for SPC/cCuO, SPC/wCuO, SPC/aCuO and SPC/eCuO modified SPC electrodes, respectively. The anodic and cathodic plots suggest diffusion-controlled processes at the electrodes.

Table 2 summarizes the electrochemical behavior of the bare electrode and the SPC modified electrodes of SPC/cCuO, SPC/wCuO, SPC/aCuO and SPC/eCuO. From this table, it is apparent that the best current response (anodic peak current, I_pa_) to the redox probe was obtained from SPC/cCuO. This was followed by the current response obtained at SPC/aCuO. Basically, the current response obtained from the electrodes followed the order SPC/cCuO > SPC/aCuO > SPC/wCuO > SPC/eCuO > bare SPC. The reversibility of the redox reaction at an electrode can be estimated from the difference of the anodic (E_pa_) and the cathodic peak potentials (E_pc_) (ΔE_p_). The closer this difference is to 59/n mV, the more reversible the process [48]. The reaction of the redox probe at the surface of the bare electrode, the electrode modified with chemically synthesized nanoparticles (NPs) and the green NPs are irreversible or quasi-reversible at best. This inference was drawn from the ΔE_p_ values of 1.04, 1.03, 1.02, 1.00 and 1.19 obtained at bare SPC, SPC/wCuO, SPC/aCuO, SPC/eCuO and SPC/cCuO, respectively. Similarly, the extent of departure of the ratio I_pa_/I_pc_ of all the electrodes from unity further confirms the irreversibility of the redox reaction at the surface of the bare electrode and the modified electrodes.
(2)$$i_{p}=\;2.69\;\times \;10^{5}\;n^{2/3}A_{eff}D^{1/2}CV^{1/2}\;$$

Using the Randles–Sevcik equation (Equation (2)) where *i_p_*, *n*, *A_eff_*, *D*, *C* and *v* represent the peak current (A), number of electrons transferred, effective surface area of the electrode (cm^2^), diffusion coefficient of the redox probe (cm^2^ s^−1^), concentration of the analyte (mol cm^−3^) and the scan rate (V s^−1^), respectively. From the plot of the anodic peak potential (I_pa_) against the square root of the scan rate (*V*^½^) (Figure 8a–d), the slope which is equal to $2.69\;\times \;10^{5}\;n^{2/3}A_{eff}D^{1/2}C$ was used to calculate *A_eff_* given the value of *D, n* and *C* for the redox probe as 7.6 × 10^−6^ cm^2^ s^−1^, 1 and 10 mM, respectively [49]. The *A_eff_* of the bare SPC, SPC/wCuO, SPC/aCuO, SPC/eCuO and SPC/cCuO were calculated as 0.69 × 10^−4^, 0.275, 0.129, 0.081 and 0.079, respectively. Again, the highest surface area was obtained for the electrode modified with cCuO while the best surface area from the electrode modified with the green CuO was obtained at SPC/aCuO. The extremely small surface area obtained for the bare SPC is an indication that the modification of the bare electrode with the green mediated CuO has affected the surface area of the electrode in no small measures. As a result, the incorporation of the CuO into other materials with large surface area could create a composite with improved surface area and electrocatalytic activity towards an analyte of interest. This possibility has precedence in literature where green mediated nanoparticles have been combined with materials such as polypyrrole [50] and graphene oxide [51,52,53] with the nanoparticles bringing forth improved electrocatalytic activity to the resultant composite. Interestingly, green-mediated CuO have also been individually used for the modification of bare glassy carbon electrode for the detection of biomolecules [51,54].

## 4. Conclusions

CuO were synthesized using a chemical and green method for a comparative study. The morphological studies revealed that the chemically synthesized CuO have similar morphology with that of the copper oxide prepared from plant extract using ethanol as solvent. Similar conclusion could be drawn from the SEM micrograph of green CuO prepared from plant extracts obtained with water and acetone as solvents. The electrochemical activity of the chemically prepared CuO towards ferrocyanide redox probe was much better than that of the green CuO. Further studies on the modification of the CuO with other conducting materials could increase their electrocatalytic activity towards the analytes of interest. On this basis, the green CuO have the prospect of application in electrochemical sensing among other industrial applications, as with the chemically synthesized CuO. 

## Figures and Tables

**Figure 1 nanomaterials-10-02502-f001:**
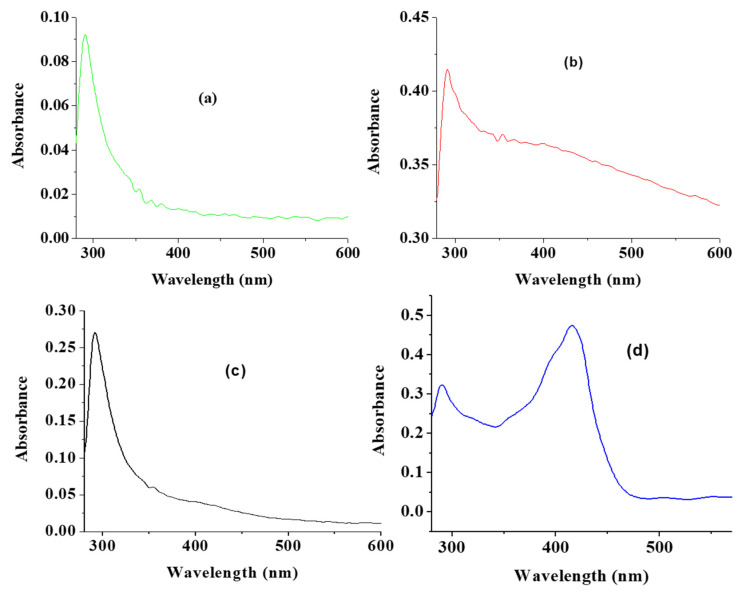
UV-vis spectra of (**a**) CuO obtained using traditional chemical means (cCuO), (**b**) CuO obtained using deionized water (wCuO), (**c**) CuO obtained using acetone (aCuO) and (**d**) CuO obtained using ethanol (eCuO).

**Figure 2 nanomaterials-10-02502-f002:**
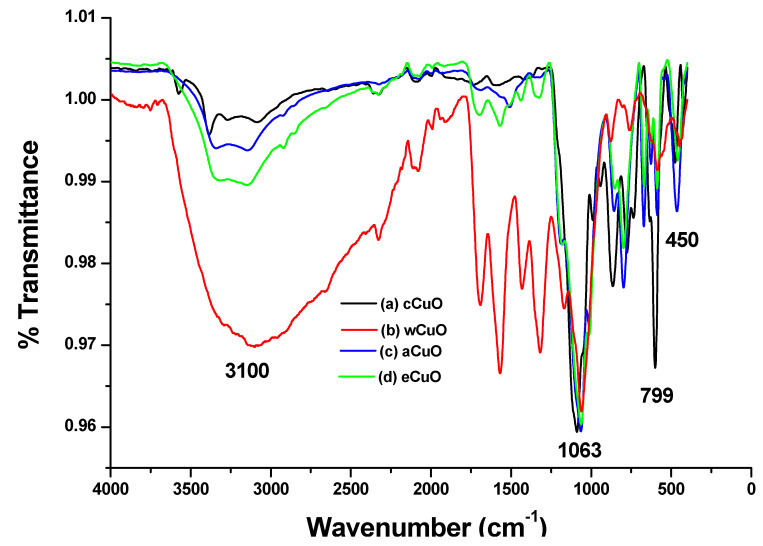
FTIR spectra of CuO from (**a**) cCuO, (**b**) wCuO, (**c**) aCuO and (**d**) eCuO.

**Figure 3 nanomaterials-10-02502-f003:**
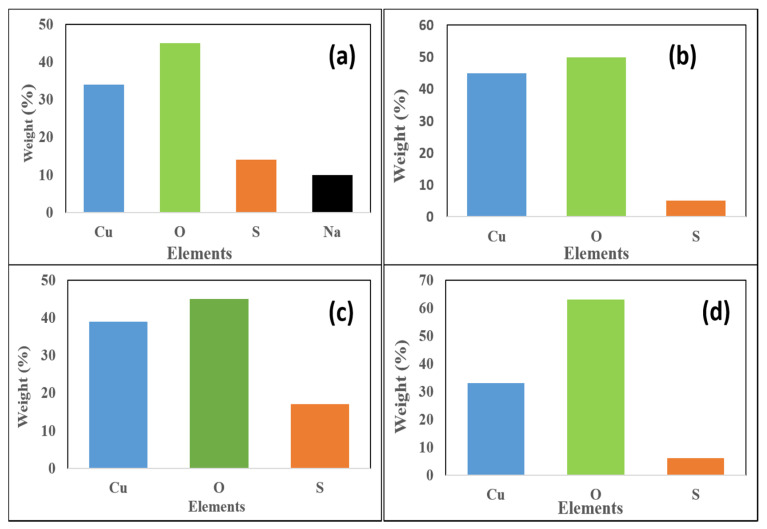
EDX plots of (**a**) cCuO, (**b**) wCuO, (**c**) aCuO and (**d**) eCuO

**Figure 4 nanomaterials-10-02502-f004:**
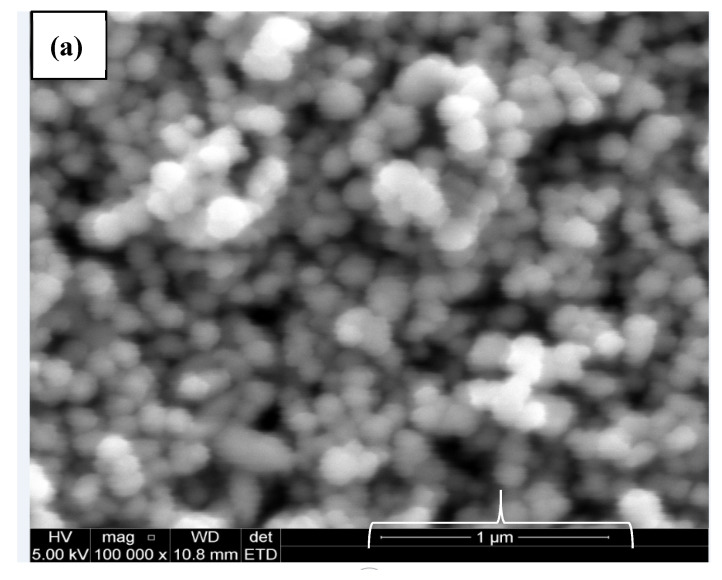
SEM images of (**a**) cCuO, (**b**) wCuO, (**c**) aCuO and (**d**) eCuO.

**Figure 5 nanomaterials-10-02502-f005:**
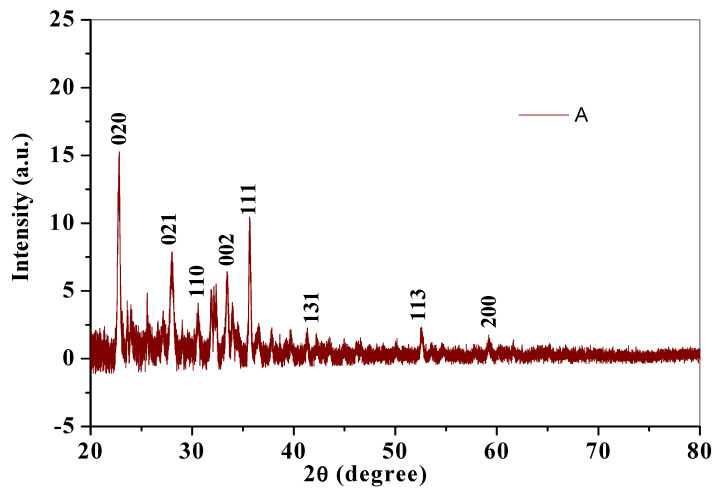
X-ray diffractograms of (**A**) cCuO and (**B**) wCuO.

**Figure 6 nanomaterials-10-02502-f006:**
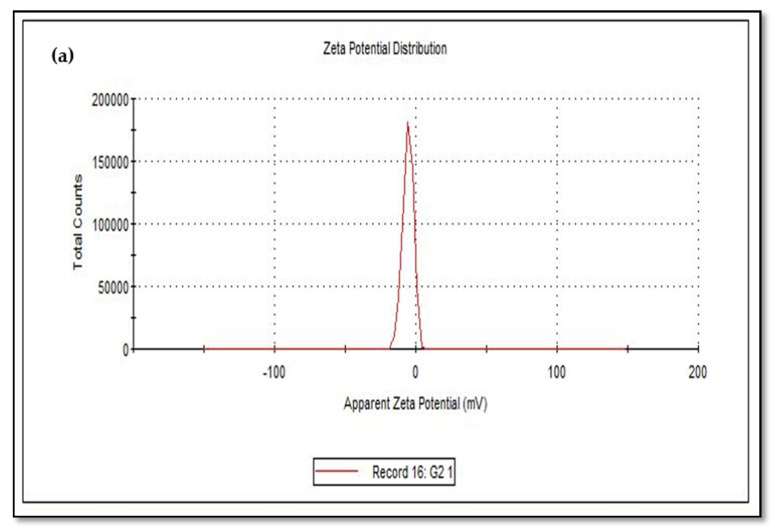
Zeta potential of (**a**) cCuO, (**b**) wCuO, (**c**) aCuO and (**d**) eCuO.

**Figure 7 nanomaterials-10-02502-f007:**
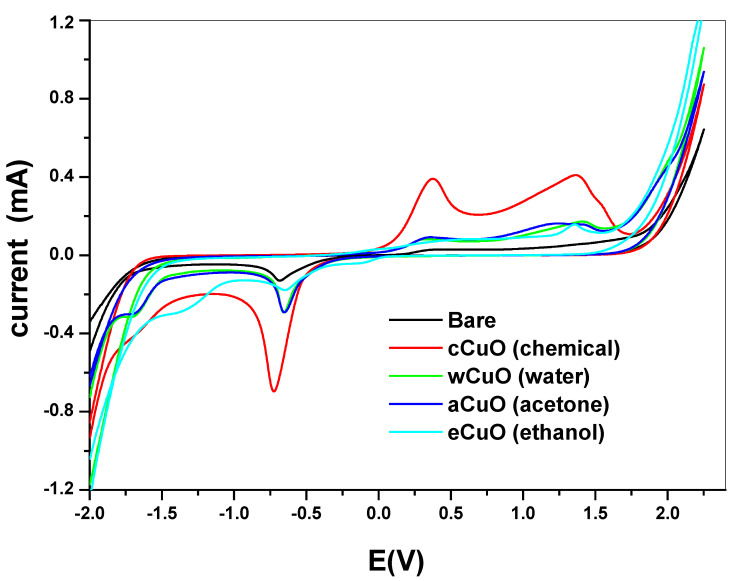
Cyclic voltammograms of the bare screen printed carbon electrode (SPC) and SPC modified with chemical and green synthesis using different solvent in 10 mM [Fe(CN)_6_]^4−^/[Fe(CN)_6_]^3−^ solution prepared in 0.1 M phosphate buffer saline (PBS) at a scan rate of 50 mV/s.

**Figure 8 nanomaterials-10-02502-f008:**
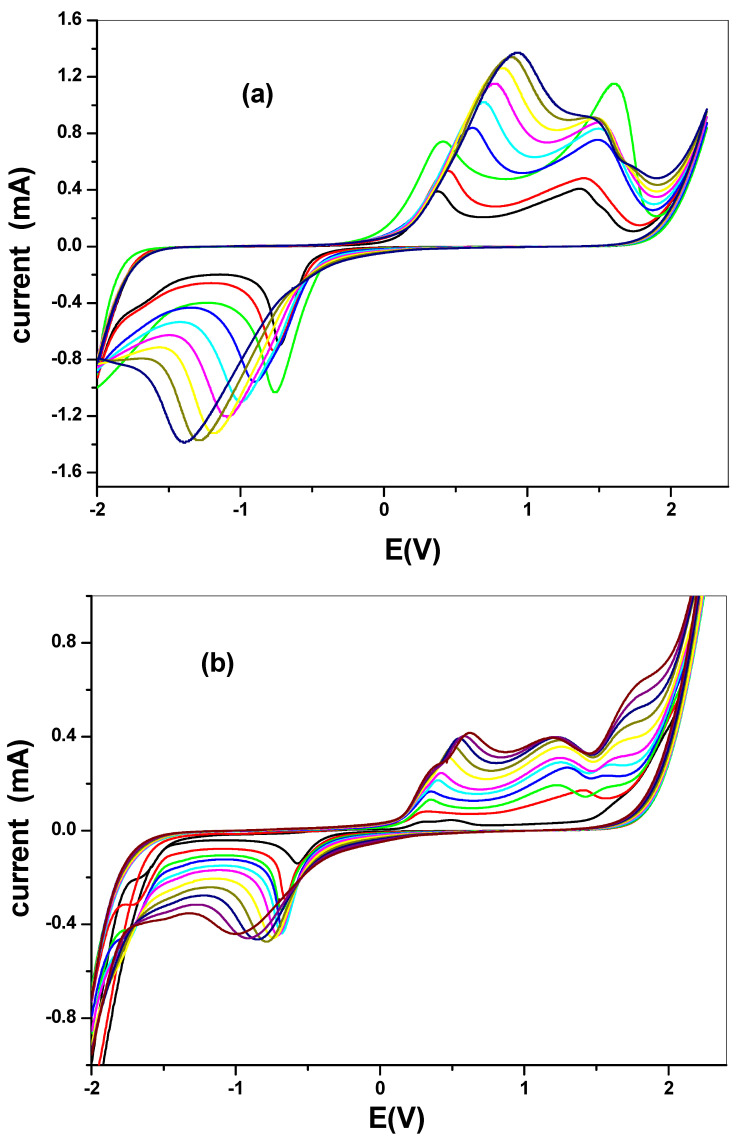
Scan rate voltammogram of (**a**) SPC/cCuO, (**b**) SPC/wCuO, (**c**) SPC/aCuO, and (**d**) SPC/eCuO nanoparticles in 10 mM [Fe(CN)_6_]^4−^/[Fe(CN)_6_]^3−^ solution prepared in 0.1 M PBS at varying scan rates (50–400 mV/s).

**Figure 9 nanomaterials-10-02502-f009:**
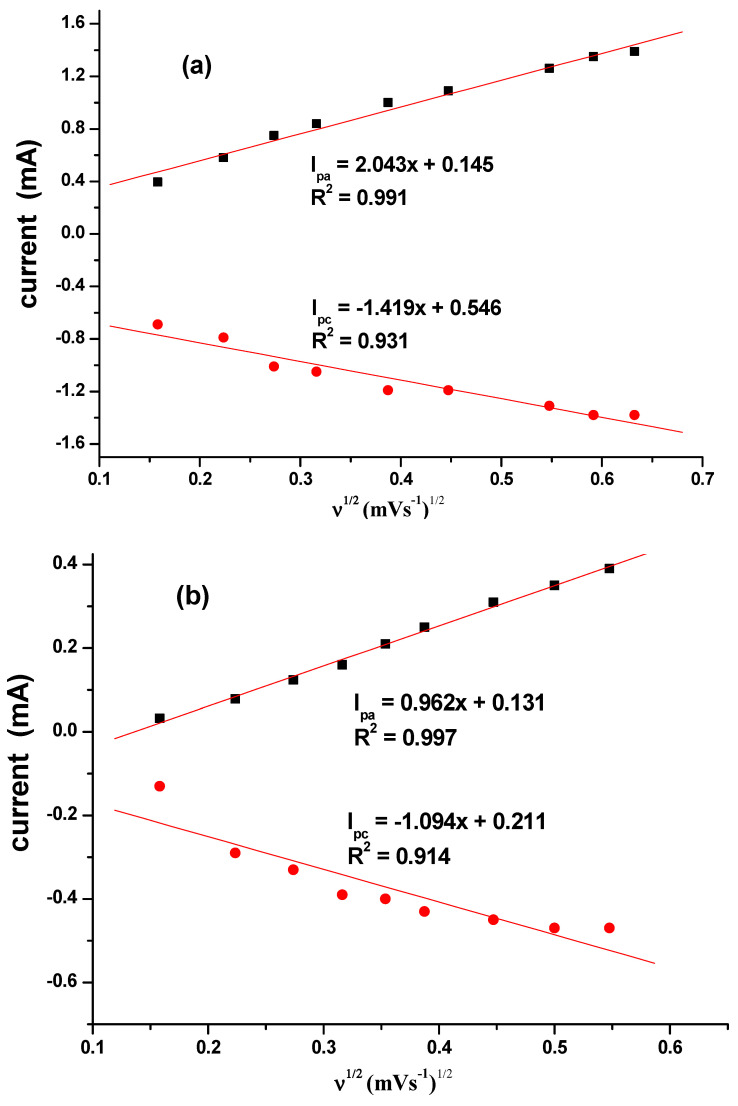
Corresponding linear plots of V^1/2^ versus currents for (**a**) SPC/cCuO, (**b**) SPC/wCuO, (**c**) SPC/aCuO and (**d**) SPC/eCuO varying scan rates (50–400 mV/s).

**Table 1 nanomaterials-10-02502-t001:** d-inter planar spacing calculations for CuONP.

Peaks (2θ)		θ		Sin θ		d (nm)	
cCuO	wCuO	cCuO	wCuO	cCuO	wCuO	cCuO	wCuO
22.85	22.91	11.42	11.45	0.1980	0.1985	0.3887	0.3879
28.00	27.96	14.00	13.98	0.2419	0.2415	0.3183	0.3187
30.58	30.60	15.29	15.30	0.2637	0.2638	0.2919	0.2637
33.42	33.44	16.71	16.72	0.2875	0.2876	0.2678	0.2875
35.66	35.67	16.83	17.83	0.2895	0.3061	0.2659	0.2895
41.43	41.34	20.71	20.67	0.3536	0.3529	0.2177	0.3536
52.45	52.69	26.22	26.35	0.4418	0.4438	0.1742	0.4420
59.16	60.18	29.58	30.05	0.4936	0.5013	0.1559	0.4939

**Table 2 nanomaterials-10-02502-t002:** Electrochemical parameters of bare SPC and electrodes modified with CuO nanoparticles.

Electrodes	E_pa_	E_pc_	I_pa_	I_pc_	A_eff_ (cm^2^)	I_pa_/I_pc_	ΔE_p_
Bare SPC	0.32	−0.72	6.87 × 10^−5^	−1.76 × 10^−4^	6.90 × 10^−5^	3.90 × 10^−1^	1.04
SPC/wCuO	0.36	−0.67	1.66 × 10^−4^	−4.12 × 10^−4^	0.12	4.03 × 10^−1^	1.03
SPC/aCuO	0.37	−0.65	1.76 × 10^-4^	−3.94 × 10^−4^	0.08	4.47 × 10^−1^	1.02
SPC/eCuO	0.34	−0.66	1.27 × 10^−4^	−2.60 × 10^−4^	0.061	4.88 × 10^−1^	1.00
SPC/cCuO	0.44	−0.75	7.32 × 10^−4^	−1.03 × 10^−3^	0.196	7.11 × 10^−1^	1.19

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
