# Peer review of "Green and Traditional Synthesis of Copper Oxide Nanoparticles—Comparative Study"

_nanomaterials, 2020, doi:10.3390/nano10122502_

Round 1

Reviewer 1 Report

CuO nanoparticles (CuO-Nps), prepared by using a conventional method (NaBH4 reduction) and a new green synthesis based on leaf extracts of Terminalia phanerophlebia, were investigated by using UV-vis, FTIR, zeta potential, SEM and EDX measurements. CV measurements, performed in presence of Fe(CN)6]3-/[Fe(CN)6]4- as redox probe, pointed out that SPC electrodes modified with CuO-Nps obtained by the chemical method exhibit the best electroactivity.

However, before publication some important issues should be carefully examined:

(i) Undoubtedly, the paper is not enough elaborated: the advantages of the new method proposed for CuO-Nps preparation are not clear mentioned; the CuO-Nps prepared by using the conventional method exhibit a better electrochemical activity than those obtained by green synthesis (explanation?); the experimental results were not statistically presented/interpreted; the weak linear dependence between the peak currents and V1/2 (Figure 7) make the information obtained from Randles-Sevcik equation less relevant.

(ii) The method used to modify the SPC electrodes with CuO-Nps, as well as the SPC/CuO-Nps modified electrodes reproducibility and stability were not described.

(iii) The whole manuscript should be carefully revised to eliminate the language/typing errors.

Author Response

AUTHORS RESPONSES TO REVIEWER 1

Reviewer 1

Comments and Suggestions for Authors

CuO nanoparticles (CuO-Nps), prepared by using a conventional method (NaBH4 reduction) and a new green synthesis based on leaf extracts of Terminalia phanerophlebia, were investigated by using UV-vis, FTIR, zeta potential, SEM and EDX measurements. CV measurements, performed in presence of Fe(CN)6]3-/[Fe(CN)6]4- as redox probe, pointed out that SPC electrodes modified with CuO-Nps obtained by the chemical method exhibit the best electroactivity.

However, before publication some important issues should be carefully examined:

(i) Undoubtedly, the paper is not enough elaborated: the advantages of the new method proposed for CuO-Nps preparation are not clear mentioned; the CuO-Nps prepared by using the conventional method exhibit a better electrochemical activity than those obtained by green synthesis (explanation?); the experimental results were not statistically presented/interpreted; the weak linear dependence between the peak currents and V1/2 (Figure 7) make the information obtained from Randles-Sevcik equation less relevant.

Authors’ response: Thank you sir for your comment. The advantages of the proposed method have been clearly presented in the manuscript. See yellow highlight on introduction section, lines 50-52, 61-64 on page 2. The reason why the conventional method exhibit better electrochemical activity than the green synthesized have been discussed better in the manuscript. See yellow highlight on lines 298-300, page 14, and section 3.4 of the manuscript. The whole section on the experimental results have been improved and statically presented. See yellow highlight of the sections of the experimental results in the manuscript.

(ii) The method used to modify the SPC electrodes with CuO-Nps, as well as the SPC/CuO-Nps modified electrodes reproducibility and stability were not described.

Authors’ response: Thanks for your comment. The methodology section on modification of SPC electrode, reproducibility and stability have been discussed as suggested. See yellow highlight on pages 4 to 5.

(iii) The whole manuscript should be carefully revised to eliminate the language/typing errors.

Authors’ response: Thank you sir. The entire manuscripts have been checked and revised carefully, eliminating all language and typographical errors.

Reviewer 2 Report

Review on article nanomaterials-997649

The article is about the demonstration of the preparation of CuO NPs by green and chemical synthesis. 

My general opinion is that this work is not suitable for communication in this form. Undeveloped on many points, the presentation of the results (quality of the figures) is under criticism.

Major problems: 

  • quality of the applied chemicals is missing. 
  • purification of the NPs: incomplete
  • zeta analyzer: incomplete 
  • the optical properties of metal NPs depends on concentration: e,g. Fig. what is the c? 
  • Fig. 3. scale bar is pale
  • Fig. 4. error?? Sd is missing. 
  • Fig. 5. terrible. print screen?? use Origin or excell . 
  • etc. In this form I reject this work. 

Author Response

AUTHORS RESPONSES TO REVIEWER 2

Reviewer 2

Comments and Suggestions for Authors

Review on article nanomaterials-997649

The article is about the demonstration of the preparation of CuO NPs by green and chemical synthesis. 

My general opinion is that this work is not suitable for communication in this form. Undeveloped on many points, the presentation of the results (quality of the figures) is under criticism.

Major problems: 

  • quality of the applied chemicals is missing. 

Authors’ response: Thanks for the comment. Qualiy of applied chemicals now added, see yellow highlight on page 4.

  • purification of the NPs: incomplete

Authors’ response: Thank you sir for the observation. The discussion on the purification of nanoparticles have been revised. See yellow highlight on on section 2.5, page 4.

zeta analyzer: incomplete 

Authors’ response: Thank you sir. The discussion of the section on zeta analysis have been completed. See yellow highlight on page 5.

  • the optical properties of metal NPs depends on concentration: e,g. Fig. what is the c? 

Authors’ response: Thanks for your observation. Sentence revised and question on c addressed. See yellow highlight section 3.1.1, page 6 of the manuscript.

  • 3. scale bar is pale

Authors’ response: Thanks. Figure 3 scale bar issue resolved by using clearer images. See Figures on page 10 of the manuscript.

  • 4. error?? Sd is missing. 

Authors’ response: Thanks. Figure 4 is now Figure 5 on the revised manuscript. Missing informations added and new spectra added also. The discussion of the result in the section has been improved. See yellow highlight on section 3.2.2, page 9 of the revised manuscript.

  • 5. terrible. print screen?? use Origin or excell . 

Authors’ response: Thanks. The Figure 5 is now Figure 6, and the graphs have been improved by using origin 7.0.

  • In this form I reject this work. 

Reviewer 3 Report

Here the authors report a “green” synthesis of copper oxide nanoparticles, which are then characterized by UV-vis, FTIR, SEM/EDX and CV. This “green” method is compared to other synthetic methods, labeled “chemical”.

- I am not comfortable with the use of “green” vs. “chemical” synthesis. Both methods are based on bottom-up syntheses that take place in solution and use reducing and stabilizing agents. One uses reducing and stabilizing agents that are plant-based, thus “green”. The other uses non plant-based reducing and stabilizing agents. However, both are “chemical”, as both involve a chemical reduction of metal ions; both also involve adsorption of a stabilizer on the metal/metal oxide surface. I suggest “green vs. traditional chemical syntheses” instead. These terms also need to be addressed though out the text.

- section 2.3.1: Are sulfate ions from CuSO4 acting as the stabilizing agent in this synthesis?

- Figure 1: Why are the peaks at ~290 nm so sharp? Also the peak absorbance is larger than 2 on all of the spectra; it doesn’t seem real?

- line 155: “3404” (units missing)

- Figure 2: Captions refer to a,a,b,c instead of a,b,c,d.

- Figure 3: The sample preparation for SEM is missing from Materials and Methods.

- Figure 4: Here we find “EDX plots” with % weight. Could we also see “EDX spectra” as proof?

Author Response

AUTHORS RESPONSES TO REVIEWER 3

Reviewer 3

Comments and Suggestions for Authors

Here the authors report a “green” synthesis of copper oxide nanoparticles, which are then characterized by UV-vis, FTIR, SEM/EDX and CV. This “green” method is compared to other synthetic methods, labeled “chemical”. 

- I am not comfortable with the use of “green” vs. “chemical” synthesis. Both methods are based on bottom-up syntheses that take place in solution and use reducing and stabilizing agents. One uses reducing and stabilizing agents that are plant-based, thus “green”. The other uses non plant-based reducing and stabilizing agents. However, both are “chemical”, as both involve a chemical reduction of metal ions; both also involve adsorption of a stabilizer on the metal/metal oxide surface. I suggest “green vs. traditional chemical syntheses” instead. These terms also need to be addressed though out the text.

Authors’ response: Thanks for your comments. Authors decided to modify the title as ‘Green and Traditional Synthesis of Copper oxide nanoparticles– Comparative Study”.

- section 2.3.1: Are sulfate ions from CuSO4 acting as the stabilizing agent in this synthesis?

Authors’ response: Thanks. yes.

- Figure 1: Why are the peaks at ~290 nm so sharp? Also the peak absorbance is larger than 2 on all of the spectra; it doesn’t seem real?

Authors’ response: Thanks for the comment. New UV spectra have been acquired by diluting the samples. See Figure 1 of the manuscript.

- line 155: “3404” (units missing)

Authors’ response: Thanks. The missing unit on value 3404 has been added. See yellow highlight on line 174, page 7 of the revised manuscript.

- Figure 2: Captions refer to a,a,b,c instead of a,b,c,d.

Authors’ response: Thanks for the brilliant observation. Captions label for Figure 2 revised accordingly. See pages 7 and 8.

- Figure 3: The sample preparation for SEM is missing from Materials and Methods.

Authors’ response: Thanks for your comments. The sample preparation protocol for SEM have been included in the materials and methods section. See yellow highlight on section 2.6, lines 134-137, on page 5 of the revised manuscript.

- Figure 4: Here we find “EDX plots” with % weight. Could we also see “EDX spectra” as proof?

Authors’ response: Thanks for the comment. One of the EDX spectra for green mediated CuO nanoparticles synthesized also in our group is hereby presented as a proof. We only acquired spectra for some selected samples while the other samples we did not receive their spectrum, only the data that shows elemental composition of our nanoparticles. We would have love to present the spectra in this study too as you have requested, but because of the global pandemic we cannot re-run the samples to generate the spectra.

Round 2

Reviewer 1 Report

The revised form of the manuscript was substantially improved. All my comments were positively considered. Consequently, I recommend its publication.

Author Response

AUTHORS RESPONSES TO REVIEWER 1b

Reviewer 1

Open Review

(x) I would not like to sign my review report

( ) I would like to sign my review report

English language and style

( ) Extensive editing of English language and style required

( ) Moderate English changes required

( ) English language and style are fine/minor spell check required

(x) I don't feel qualified to judge about the English language and style

Yes      Can be improved     Must be improved   Not applicable

Does the introduction provide sufficient background and include all relevant references?

( )         (x)        ( )         ( )

Is the research design appropriate?

( )         (x)        ( )         ( )

Are the methods adequately described?

( )         (x)        ( )         ( )

Are the results clearly presented?

( )         (x)        ( )         ( )

Are the conclusions supported by the results?

( )         (x)        ( )         ( )

Comments and Suggestions for Authors

The revised form of the manuscript was substantially improved. All my comments were positively considered. Consequently, I recommend its publication.

Authors’ responses

Thanks for your comments. All suggested areas of improvement on the manuscript have been addressed and highlighted yellow.

Reviewer 2 Report

The presentation of the figures is still terrible.

e.g. the font style is different for fig. 1-8. 

The quality is the DLS curves is still as print screen. 

Please make a table for data presentation and export the data from the software and edit again. see this MDPI article as template: Pharmaceutics 2019, 11, 0357; doi:10.3390/pharmaceutics11070357 (FT-IR and DLS curves. ) 

Major revision is required. 

Author Response

AUTHORS RESPONSES TO REVIEWER 2b

Reviewer 2

The presentation of the figures is still terrible.

e.g. the font style is different for fig. 1-8.

Authors’ response: Thanks. All figures mentioned above have been checked and redrawn with consistence font style. See Figures on pages 7 – 15 of the manuscript.

The quality is the DLS curves is still as print screen.

Please make a table for data presentation and export the data from the software and edit again. see this MDPI article as template: Pharmaceutics 2019, 11, 0357; doi:10.3390/pharmaceutics11070357 (FT-IR and DLS curves. )

Authors’ response: Thanks for your comments. The MDPI article suggested as template was used to edit the FTIR figures.
